# Instance-dependent Approximation Guarantees for Lipschitz Approximators, Application to Scientific Machine Learning

## Abstract

Despite widespread adoption, Machine Learning models remain data-driven and lack exploitable theoretical guarantees on their approximation error. This limitation hinders their use for critical applications. In this paper, we show how to leverage the Lipschitz property for Lipschitz approximations, i.e., ML models that are Lipschitz continuous, to establish strict post-training — instance dependent – generalization error bounds given a set of validation points. We focus on the test case domain of ML for scientific computing called Scientific Machine Learning (SciML), where ML models are increasingly used but miss the theoretical approximation guarantees of classical scientific computing simulation schemes. We first show how to derive error bounds using Voronoï diagrams for a Lipschitz approximator trained to learn a $K$-Lipschitz function by taking advantage of the mesh-like structure of learning points. Second, we cast upper bounding as an optimization problem and use certified Deterministic Optimistic Optimization (introduced in Bachoc et al. (2021)) and certified Voronoï Optimistic Optimization (that we design based on the non-certified version in Kim et al. (2020)), to achieve tighter error bounds. The code is made available at https://anonymous.4open.science/r/lipschitz_bounds_doo-7FDF.

## 1 Introduction

Machine Learning (ML) has become ubiquitous due to its remarkable ability to learn from data. However, ML models are fundamentally statistical and trained to minimize empirical error. This characteristic results in a lack of theoretical guarantees on their approximation error, which poses significant concerns for applications requiring high reliability, such as safety-critical and AI-for-science applications.

Previous works have attempted to provide bounds on the approximation error. However, these bounds either are probabilistic Haussler (1992) or applies to the hypothesis space of the model of interest, prior to the training, and do not consider the learning phase and the actual error of the model on learning data points post-training Bartlett & Mendelson (2002); Bartlett (1996); Bousquet & Elisseeff (2002); Bartlett et al. (2017); Jacot et al. (2018). A large body of methods based on formal verification consider the trained model, but only provide local guarantees around a given input Katz et al. (2017); Ehlers (2017); Vidot et al. (2022); Zhang et al. (2018).

In this paper, we address these gaps in the case where ML models are Lipschitz continuous - a property shared by some classes of ML models such as Lipschitz neural networks or Gaussian Processes. We propose post-training generalization error bounds for Lipschitz approximators. These instance-dependent bounds leverage all available information for the learning task, as they depend on the trained ML model, the learning data points, and the validation data points.

Although the presented bounds are general and apply to any Lipschitz approximators trained to learn a Lipschitz function, our primary application domain is Scientific Machine Learning (SciML), i.e., ML for scientific computing, where ML models are increasingly employed but often lack the theoretical approximation guarantees provided by classical scientific computing simulation schemes. Some works have derived generalization bounds for such test cases based on the knowledge of the PDE underlying the function to learn. However, the bounds either are theoretical Kovachki et al. (2021); Lee & Shin (2024), or only applicable for

one instance of PDE De Ryck & Mishra (2022); Mishra & Molinaro (2023) are not adapted to the setting where the parameters of the PDE change in the dataset - which is quite common, for instance, the recent NeurIPS 2024 competition ML4CFD Yagoubi et al..

We first demonstrate how to derive error bounds using Voronoï diagrams for a $K_g$-Lipschitz approximation $g$ trained to learn a $K_f$-Lipschitz function $f$. As we shall see, this approach is cost expensive because the Voronoï diagram's construction has an exponential complexity with respect to the dimension and the number of points Aurenhammer (1991). To alleviate this complexity, we take advantage of the mesh-like structure of learning points, which is common in scientific computing due to the presence of spatio-temporal inputs. Subsequently, we reformulate the problem of upper bounding as an optimization problem. We employ certified Deterministic Optimistic Optimization (introduced in Bachoc et al. (2021)). Moreover, we introduce certified Voronoï Optimistic Optimization (designed based on the non-certified version in Kim et al. (2020)).

Our contributions can be summarized as follows:

- We propose strict post-training generalization error bounds for Lipschitz approximators, leveraging the Lipschitz property of the approximator and the learning data points using Voronoï diagrams.

- We demonstrate how to take advantage of the mesh-like structure of some input's dimension to alleviate the computational complexity of Voronoï diagrams.

- We cast the problem of upper bounding as an optimization problem to enable computing bounds using certified optimization algorithms.

- We employ certified Deterministic Optimistic Optimization (introduced in Bachoc et al. (2021)) and introduce certified Voronoï Optimistic Optimization to achieve tighter error bounds.

After a review of related works on generalization error bounds for machine learning models in Section 2, Section 3 introduces the theoretical background and main idea of bounding the generalization error using the Lipschitz property. Then, in Section 4, we present an approach to evaluate the bound using Voronoï diagrams and discuss its computational aspects. Section 5 reformulates the problem of upper bounding as a certified deterministic optimistic optimization problem and introduces c-DOO and c-VOO. Finally, Section 6 discusses the perspectives and limitations of our approach and suggests potential future research directions.

## 2    Related Works

The goal of providing generalization bounds for ML models has already been thoroughly explored and is still a very active area of research. Over the years, researchers have proposed various theoretical frameworks to explain generalization and come up with generalization bounds. For instance, the Vapnik-Chervonenkis (VC) dimension Bartlett (1996), Rademacher Complexity Bartlett & Mendelson (2002), stability analysis Bousquet & Elisseeff (2002), or neural tangent kernel theory Jacot et al. (2018) all aims to provide insights into generalization capabilities of neural networks. Still, they mostly focus on the hypothesis space of a given ML model and the distribution of the learning points and do not consider the information available after the training, i.e. the learning points at hand and the trained model. PAC-Bayes theory (see Haussler (1992) and stemming references) also provides generalization bounds, but these bounds are probabilistic. Furthermore, all these bounds are computed to control the $L_2$ error. In this work, we explore *instance-dependant* exact worst-case generalization bounds, i.e., deterministic bounds that can be computed using the trained ML model and the learning dataset. Our goal is to provide operative bounds for AI applications such as certification and worst-case analysis.

Another family of methods seeks to leverage data points and the ML model once trained, namely formal methods. These methods, used for ML robustness verification, aim to provide local guarantees on the model's behavior in the neighborhood of a given input. They are often based on optimization problems, such as robust optimization, mixed-integer linear programming, Satisfiability Modulo Theories (SMT) solvers Katz et al. (2017); Ehlers (2017); Vidot et al. (2022); Zhang et al. (2018); Wong et al. (2018), interval bound

propagation Singh et al. (2019); Zhang et al. (2022), or neural networks relaxation De Palma et al. (2023); Xu et al. (2020) but suffer from their complexity, thereby limiting their applicability to large neural networks. In this work, we propose a method to provide generalization bounds for any Lipschitz approximation of arbitrary size that is not limited to piecewise linear models.

Finally, in the specific case of Scientific Machine Learning, generalization error has been studied, but results are mainly theoretical Kovachki et al. (2021); Lee & Shin (2024) or leverages PDE-specific information Mishra & Molinaro (2023); De Ryck & Mishra (2022). In this work, our goal is to provide bounds for any class of learning problem, regardless of the underlying PDE, and to unlock generalization bounds for use cases that encompass different instances of PDE solutions, e.g., when PDE's parameters or boundary conditions involved in the simulation change within the dataset (as illustrated in NeurIPS 2024 competition ML4CFD Yagoubi et al.).

Our approach draws inspiration from Lipschitz optimization and interpolation. In particular, Beliakov (2006) propose non-parametric optimal Lipschitz interpolators, and DOO Munos (2011) together with a line of bandit algorithms such as more recent LIPO Malherbe & Vayatis (2017), HOO Bubeck et al. (2011), or Zooming algorithm Kleinberg et al. (2019), optimize a function with the only knowledge of its smoothless (i.e. its Lipschitz constant). In this work, we use the more recent Bachoc et al. (2021), inspired from Piyavskii (1972); Shubert (1972) and build on Kim et al. (2020) to provide certified Lipschitz bounds for the generalization error of Lipschitz approximators.

## 3 A Bound for the Error of Lipschitz Approximators

### 3.1 Setting

Let's consider the following learning problem. Let $f : \mathbb{X} \subset \mathbb{R}^d \to \mathbb{R}$ be a function we want to approximate using a machine learning model $g$. We consider a set of $n$ learning points $\left\{(x_i, f(x_i))\right\}_{i=1}^n$, where $\{x_i\}_{i=1}^n$ are sampled from a probability distribution $P_x$. The approximation is achieved by minimizing a cost function evaluated on these points.

The quality of the approximation is assessed by the generalization error $J_g$, which we define as

$$J_g = \|f - g\|,$$

where $\|.\|$ is a norm that is usually chosen as the $L_1$ norm, $\int_{x \in \mathbb{X}} |f(x) - g(x)| dP_x$, or the $L_2$ norm $\int_{x \in \mathbb{X}} (f(x) - g(x))^2 dP_x$. In this paper, we are interested in worst-case approximation guarantees, so we focus on the $L_\infty$ norm:

$$J_g = \max_{x \in \mathbb{X}} |f(x) - g(x)|.$$

The goal of this paper is to derive bounds on $J_g$ in the case where $f$ and $g$ enjoy the Lipschitz property defined as following:

**Definition 3.1** (Lipschitz Property). A function $f : \mathbb{X} \to \mathbb{R}$ satisfies the Lipschitz property for a norm $\|.\|$ when there exists a constant $K_f \in \mathbb{R}^+$ such as $\forall x, y \in \mathbb{X}^2$,

$$|f(x) - f(y)| \leq K_f \|x - y\|.$$

In that case, $f$ is said $K_f$-Lipschitz. In the following, we only consider the Euclidian norm.

To achieve this goal, we will suppose that there exists $K_f$ such that $f$ is $K_f$-Lipschitz, which is a mild hypothesis. As for $g$, standard ML models such as neural networks, gaussian processes, or polynomials (when $\mathbb{X}$ is bounded) naturally enjoy the Lipschitz property, whose constant is denoted $K_g$. However, it can be challenging to evaluate $K_f$ and $K_g$.

### 3.2 Evaluating $K_f$ and $K_g$

**Evaluating $K_f$**   Knowing the exact Lipschitz constant of a function is not trivial. For black-box functions $f$, i.e., functions we don't have access to, either because data comes from the real world or was generated using a complex procedure (like experiments or simulations), we can often only approximate it.

Evaluating a function's Lipschitz constant is a topic on its own (see e.g., Wood & Zhang (1996); Weng et al. (2018); Calliess (2015); Fazlyab et al. (2019)) but for the sake of simplicity, here we approximate $K_f$ as:

$$\widehat{K_f} = \max_{i,j\in\{1,\dots,n\}^2} \frac{|f(x_i) - f(x_j)|}{\|x_i - x_j\|} \tag{1}$$

In that case, $\widehat{K_f}$ is a lower bound for $K_f$. However, we can see it as the Lipschitz constant of a piecewise linear function interpolating $\{(x_i, f(x_i))\}_{i=1}^n$. Hence, even if it is not the actual Lipchitz constant of $f$, using it to compute bounds for $g$ ensures that it does not behave unsteadily between learning points.

Note that for some applications, $K_f$ may be known thanks to the knowledge of $f$ Bunin & François (2017). In the following, for simplicity, we directly use the notation $K_f$, regardless of whether it is exact or approximated.

**Evaluating $K_g$**  As for $K_g$, depending on the ML model we use, it can be more or less complicated to evaluate. In Lederer et al. (2019), the authors introduce a method to compute the Lipschitz constant of Gaussian processes. For polynomials, it corresponds to the maximum of the norm of the model's gradient on $\mathbb{X}$, which involves a non-convex optimization problem to find. It is even more challenging for neural networks because evaluating their Lipschitz constant is known to be an NP-hard problem Virmaux & Scaman (2018). Still, a body of works intends either to provide tractable upper bounds Pauli et al. (2021); Huang et al. (2021); Bungert et al. (2021), or to build neural networks whose Lipschitz constant is enforced by construction Anil et al. (2019); Serrurier et al. (2021); Wang & Manchester (2023). As we shall see in the following, our bounds directly depends on the Lipschitz constant of the neural network. Hence, we choose the class of models whose Lipschitz constant is known by construction and focus on Norm-Preserving Neural Networks Anil et al. (2019); Serrurier et al. (2021).

*Remark* 3.2. Note that neural networks based on self-attention are not Lipschitz. However, some works design orthogonal attention, which solves this issue Xiao et al. (2024).

## 3.3 Bounding Using Lipschitz Property

The objective of the paper is to upper bound $J_g$, i.e., the function $e : \mathbb{X} \to \mathbb{R}$ such as $e(x) = |f(x) - g(x)|$. Throughout the paper, we will rely on the following proposition:

**Proposition 3.3.** *Let $e$ be defined as above. Then, $\forall x, y \in \mathbb{X}^2$,*

$$e(y) \le e(x) + (K_f + K_g)\|x - y\|. \tag{2}$$

This proposition straightforwardly stems from the Lipschitz property of $f$ and $g$. Proposition 3.3 will be used to upper bound $e$ on $\mathbb{X}$ because $\forall y \in \mathbb{X}$, it allows bounding $e(y)$ with an evaluation of $e$ on arbitrary $x$, which is already known if $x$ is chosen from the learning data points, and $\|x - y\|$ which can be trivially computed.

## 4 Evaluating the Bound Using Voronoï diagram

In this section, we investigate a first way of using 3.3 to achieve error bounds, based on partitioning $\mathbb{X}$ around $\{x_i\}_{i=1}^n$. Formally, we rely on the following proposition:

**Proposition 4.1.** *Let's consider a partition $\{\mathbb{S}_i\}_{i=1}^n$ of $\mathbb{X}$, where each element of the partition is called a cell, such that $\mathbb{X} = \bigcup_{i=1}^n \mathbb{S}_i$, $\bigcap_{i=1}^n \mathbb{S}_i = \varnothing$ and $\forall i \in \{1, \dots, n\}, x_i \in \mathbb{S}_i$. Let $r(x_i) = \max_{x \in \mathbb{S}_i} \|x - x_i\|$. Then,*

$$J_g \le \max_{i \in \{1,\dots,n\}} \left\{ e(x_i) + (K_f + K_g)r(x_i) \right\}. \tag{3}$$

This proposition is illustrated in Figure 1, and the proof is left in Appendix A.1. The question now arises as to how to choose a partition cleverly. To guide this choice, let's consider the following proposition:

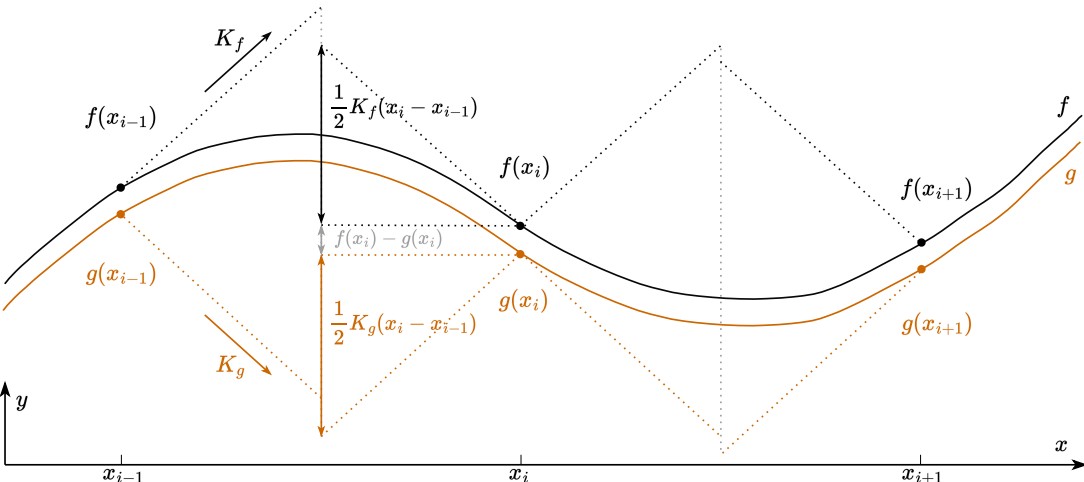

Figure 1: Illustration of Proposition 4.1. When $f$ and $g$ are Lipschitz, we can bound their variation inside each element of the partition and therefore bound $f - g$. In that case, $\mathbb{S}_i = [x_i - \frac{|x_i - x_{i-1}|}{2}, x_i + \frac{|x_{i+1} - x_i|}{2}]$.

**Proposition 4.2.** *Let $N$ be the nearest neighbor map built on $\{x_i\}_{i=1}^n$, i.e. $N(x) = \underset{\{x_i\}_{i=1}^n}{\arg\min} \|x - x_i\|$. Note that $N(x)$ is a subset of $\{x_i\}_{i=1}^n$. Then, $\forall x \in \mathbb{X}$ and $\forall x' \in N(x)$*

$$e(x) \le e(x') + (K_f + K_g)\|x - x'\|,$$

Proposition 4.2 allows naturally defining a partition $\{\mathbb{S}_i\}_{i=1}^n$, which turns out to be the Voronoï diagram of $\{x_i\}_{i=1}^n$. We recall its definition in Definition 4.3.

**Definition 4.3** (Voronoï diagram). Let $\{x_i\}_{i=1}^n$ be a set of $n$ points in $\mathbb{X}$. The Voronoï diagram of $\{x_i\}_{i=1}^n$ is the partition of $\mathbb{X}$ into $n$ cells $\{\mathbb{V}(x_i)\}_{i=1}^n$ such that $\forall x \in \mathbb{V}(x_i)$, $x_i$ is the nearest neighbor of $x$.

Proposition 4.2 justifies using Voronoï diagrams because $\forall i, x \in \mathbb{V}(x_i) \Leftrightarrow x_i \in N(x)$. Hence, to upper bound $J_g$ using Voronoï diagrams, we first have to compute the Voronoï diagram, and for each Voronoï cell $\mathbb{V}(x_i)$, compute $r(x_i) = \underset{x \in \mathbb{V}(x_i)}{\max}\|x - x_i\|$. The Voronoï cells are convex sets Aurenhammer (1991), so the farthest point from $x_i$ belonging to $\mathbb{V}(x_i)$ is among the set of nodes of this cell. Let $\mathbb{N}_i$ be the set of nodes of $\mathbb{V}(x_i)$, we can obtain $r(x_i)$ using:

$$r(x_i) = \underset{x \in \mathbb{N}_i}{\max}\|x - x_i\| \tag{4}$$

And then upper bound $J_g$ using equation 3.

*Remark* 4.4. All the bounds presented in this paper can be computed with the full training dataset and do not require to split the dataset into training and validation sets, as usually done in ML for statistical error estimation. This is a significant advantage because it allows us to use all the available information to compute the bounds.

The remaining of this section studies three specific cases depending on the structure of $\{x_i\}_{i=1}^n$. First, as a warm-up example, we study the case where it is structured as a homogeneous grid where Voronoï cells reduce to hyperrectangles. Then, we study the more realistic case where data points are randomly sampled, and we have to compute the complete Voronoï diagram. Finally, we present a result specific to cases where the data points are structured as a mesh for some dimensions and randomly sampled for others. This case often occurs in practice in SciML, for instance, when approximating some PDE solutions using neural implicit representations Sitzmann et al. (2020a).

### 4.1 Warm up: Data structured as a Grid

Let's first consider that $\{x_i\}_{i=1}^n$ is structured as a grid. First, we formally define such a structure.

**Definition 4.5** (Grid structured data). Suppose that $\mathbb{X} = [0,1]^d$. We say that data $\{x_i\}_{i=1}^n$ is structured as a grid of parameters $\{p_l\}_{l=1}^d$, $p_l \in \mathbb{N}$ when $\forall i \in \{1,\dots,n\}$, there exist $\{i_l\}_{l=1}^d$, $i_l \in \{0,\dots,p_l-1\}$ such that $x_i = \left(\frac{i_1}{p_1-1}, \dots, \frac{i_d}{p_d-1}\right)$.

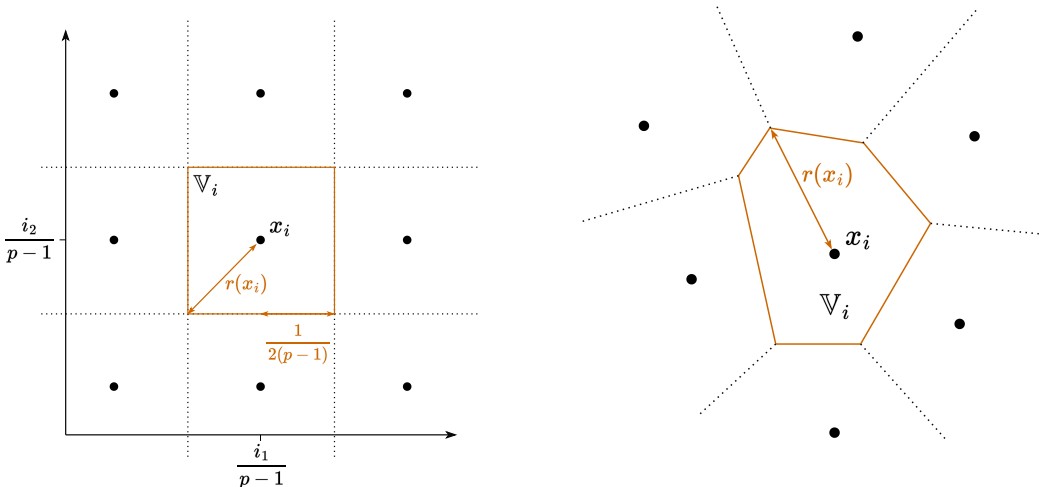

Figure 2: **Left:** Illustration of a grid structured data in 2D. **Right:** Illusttration of a Voronoï cell in 2D.

As an example, suppose that $\mathbb{X} = [0,1]^2$. We note $p = \sqrt{n}$, which is an integer and $\forall i \in \{1,\dots,n\}$, we can express $x_i$ with some $i_1, i_2 \in \{0,\dots,p-1\}^2$ such as $x_i = \left(\frac{i_1}{p-1}, \frac{i_2}{p-1}\right)$. This case is illustrated in figure 2.

$$\mathbb{V}_i = \begin{cases} \left[\dfrac{i_1}{p-1}, \dfrac{i_1}{p-1} + \dfrac{1}{2(p-1)}\right] \times \left[\dfrac{i_2}{p-1}, \dfrac{i_2}{p-1} + \dfrac{1}{2(p-1)}\right] & \text{for } i_1, i_2 = 0 \\[2ex] \left[\dfrac{i_1}{p-1} - \dfrac{1}{2(p-1)}, \dfrac{i_1}{p-1}\right] \times \left[\dfrac{i_2}{p-1} - \dfrac{1}{2(p-1)}, \dfrac{i_2}{p-1}\right] & \text{for } i_1, i_2 = p-1 \\[2ex] \left[\dfrac{i_1}{p-1} - \dfrac{1}{2(p-1)}, \dfrac{i_1}{p-1} + \dfrac{1}{2(p-1)}\right] \times \left[\dfrac{i_2}{p-1} - \dfrac{1}{2(p-1)}, \dfrac{i_2}{p-1} + \dfrac{1}{2(p-1)}\right] & \text{otherwise.} \end{cases}$$

Hence, it is a hypercube and $r(x_i) = \frac{\sqrt{2}}{2(p-1)}$, the half diagonal of $\mathbb{V}_i$. Then, we have that

$$J_g \leq \max_{i \in \{1,\dots,n\}} e(x_i) + (K_f + K_g)\frac{\sqrt{2}}{2(p-1)}.$$

In the general case, $\mathbb{V}_i$ is an hyperrectangle and $r(x_0) = \cdots = r(x_n) = \frac{1}{2}\sqrt{\sum_{l=1}^d \frac{1}{(p_l-1)^2}}$ (the half diagonal of a hyperrectangle of size $\frac{1}{p_1-1}, \dots, \frac{1}{p_d-1}$). We can then state the same result for any grid in dimension $d$:

$$J_g \leq \max_{i \in \{1,\dots,n\}} e(x_i) + \frac{1}{2}(K_f + K_g)\sqrt{\sum_{l=1}^d \frac{1}{(p_l-1)^2}}. \tag{5}$$

The utility of this special case, where data is structured as a grid, is twofold. First, it helps to grasp the underlying idea of partitioning $\mathbb{X}$ to upper bound $J_g$. Second, this case where the Voronoï diagram reduces to a grid, so $r(x_i)$ is obtained analytically, will be used in more complex approaches in the following sections.

## 4.2 Random Data Points

When data points are structured as a grid, the upper bound can be easily computed using equation 5. Moreover, computing the bound is cheap because it only involves evaluating the error function on learning or validation points, which is always done in ML for validation purposes. However, this data structure is very constraining for several reasons. First, in ML, data points are rarely structured as a grid. Second, even when we have control over $f$ (e.g., in SciML), grids suffer from the curse of dimensionality, and we may favor Quasi-Monte Carlo sampling of learning data points for better space coverage.

In more realistic cases where data is structured randomly, i.e., $\{x_i\}_{i=1}^n$ are samples of any probability distribution $P_x$, we have to compute the full Voronoï diagram of $\{x_i\}_{i=1}^n$ to obtain the bound. It can be achieved by using existing libraries such as scipy Virtanen et al. (2019), which compute Voronoï diagrams and return the edges and the nodes of each cell. In turn, knowing that the farthest point from a Voronoï cell's center is one of its nodes, we can use equation 4 to compute $\{r(x_i)\}_{i=1}^n$ and then the upper bound with equation 3. An illustration of $r(x_i)$ is given in Figure 2.

Using complete Voronoï diagrams alleviates constraints related to grids but brings another layer of complexity. Indeed, computing a Voronoï diagram is of exponential algorithmic complexity on $n$ and $d$ Aurenhammer (1991). Hence, as we shall see in Section 4.4, even for problems of moderate dimension ($d = 6$ in the presented test case), Voronoï diagrams are not affordable. There are some perspectives on unlocking Voronoï diagrams for higher dimensions that we leave for future work; see Section 6. In the meantime, we introduce one such technique for a specific case when data is structured as a grid for some dimensions and randomly for others.

## 4.3 Mixed-random-Grid Data Points

In some cases, data may be structured as a grid for some dimensions and randomly for others. Such a case occurs when using implicit neural representation in SciML, which is the main building block of many approaches Serrano et al. (2023); Catalani et al. (2024); Raissi et al. (2019); Sitzmann et al. (2020b). First, let us formally define "mixed-random-grid data points."

**Definition 4.6** (Mixed-random-grid structured data). Let $\mathbb{J} \subset \{1, \ldots, d\}$ and $\mathbb{K} := \{1, \ldots, d\} \setminus \mathbb{J}$. Let $\mathbb{X}_{\mathbb{J}}$ and $\mathbb{X}_{\mathbb{K}}$ be the spaces built with the dimensions of $\mathbb{X}$ indexed by elements of $\mathbb{J}$ and $\mathbb{K}$, respectively. Let's note $x_{i,\mathbb{J}}$ and $x_{i,\mathbb{K}}$ the vectors built from $x_i$ by selecting its dimensions indexed by elements of $\mathbb{J}$ and $\mathbb{K}$. We say that data $\{x_i\}_{i=1}^n$ is structured as mixed-random-grid when $\{x_{i,\mathbb{J}}\}_{i=1}^n$ is structured as a grid and $\{x_{i,\mathbb{K}}\}_{i=1}^n$ is randomly sampled from any distribution $P_x$ on $\mathbb{X}_{\mathbb{K}}$.

**Proposition 4.7.** *Let $\{x_i\}_{i=1}^n$ be structured as a Mixed-random-grid as defined in Definition 4.6, with a grid of parameters $\{p_j\}_{j \in \mathbb{J}}$. Let $\mathbb{V} = \{\mathbb{V}_{i,\mathbb{K}}\}_{i=1}^n$ be the Voronoï diagram of $\{x_{i,\mathbb{K}}\}_{i=1}^n$. Then,*

$$r(x_i) = \sqrt{r(x_{i,\mathbb{K}})^2 + \sum_{j \in \mathbb{J}} \frac{1}{4(p_j - 1)^2}}. \tag{6}$$

The proof is left in Appendix A.2. Proposition 4.7 has strong implications. First, we no longer need to compute a Voronoï diagram in dimension $d$ but only in dimension $d - |\mathbb{J}|$, which is a significant computational gain since the complexity of Voronoï diagrams is exponential in $d$. Second, since we only use $\{x_{i,k \in \mathbb{K}}\}_{i=1}^n$ to build $\mathbb{V}$, the number of points used to construct the diagram is $n / \prod_{j \in \mathbb{J}} p_j$ instead of $n$, because we only need to compute $r(x_{i,\mathbb{K}})$. This is another significant gain since the construction of $\mathbb{V}$ is also exponential in $n$. Hence, we can compute the bound on $J_g$ using equation 3 with $r(x_i)$ computed using equation 6 dramatically more efficiently. These computational gains are illustrated in practice on realistic test cases in Section 4.4.

Proposition 4.7 only applies when data is mixed-random-grid structured. Such a case is typical in SciML, where data comes from simulations often conducted on regular and structured meshes, with $\mathbb{J}$ corresponding to time and space coordinates. In such cases, we can then remove up to $|\mathbb{J}| = 4$, dimensions ($x, y, z$ and $t$) for the Voronoï diagram computation, which can make the calculation tractable for dimensions' values for which it would not be affordable at all.

## 4.4 Experiments

We now present experiments that illustrate the different bounds obtained using Voronoï diagrams. We consider four test cases: sinus function in two dimensions, Holder table function, heat diffusion, and flow in a pipe. The first two are simple toy functions that we use to investigate computational aspects in practice, and the last two are more complex problems that are (simple) SciML applications.

The test cases are detailed in Appendix B. Still, we give a brief overview of the test cases here, as well as illustrations in Figure 3.

- **Sinus function**: $f : x, y \mapsto \sin(x)\sin(y)$, $x, y \in [-5, 5]^2$.

- **Holder table function**: $f : x, y \mapsto \left|sin(x)cos(y)\exp\left(\left|1 - \frac{\sqrt{x^2+y^2}}{\pi}\right|\right)\right|$, $x, y \in [-5, 5]^2$.

- **Heat diffusion**: $f : \mathbb{X} \subset \mathbb{R}^6 \to \mathbb{R}$ as the stationary solution of the heat equation in two dimensions with 4 Dirichlet boundary conditions. The dimensions in $\mathbb{X}$ correspond to the two spatial dimensions and the four boundary conditions. The function $f$ returns the temperature over the domain. We run 5000 simulations on a $32 \times 32$ grid for a total of $n = 512 \times 10^4$ learning points.

- **Flow in a pipe**: $f : \mathbb{X} \subset \mathbb{R}^4 \to \mathbb{R}$ as the stationary solution of a vicious flow in a pipe. The dimensions in $\mathbb{X}$ correspond to the two spatial dimensions, the fluid viscosity, and the upstream speed. The function $f$ returns the fluid horizontal speed in the pipe. We run 2000 simulations on a $50 \times 32$ grid for a total of $n = 320 \times 10^4$ learning points.

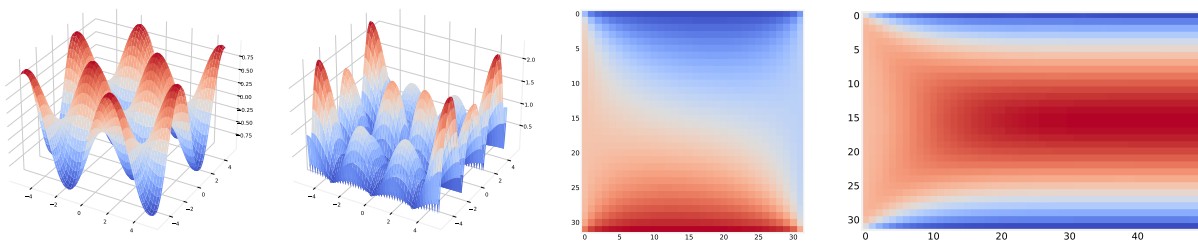

Figure 3: Illustration of the four test cases. From left to right: sinus function, Holder table function, heat diffusion, and flow-in-a-pipe. For the two latter, we only visualize the result of one simulation, corresponding to fixed boundary conditions (for heat diffusion) and upstream speed and viscosity (for flow-in-a-pipe).

For the learning part, we use a Norm-Preserving NN backbone Serrurier et al. (2021) and add an unconstrained linear layer on top. Hence, its Lipschitz constant $K_g$ can be directly obtained by computing the spectral norm of the last layer's weight matrix. We use the known Lipschitz constant of the neural network $K_g$ and estimate $K_f$ using equation 1 to compute the bounds on $J_g$ for the different test cases. Note that the learning falls under the Implicit Neural Representation formulation.

**Sinus and Holder table**    For the first two test cases, we compute the bound using the Voronoï diagrams only. To investigate the computational complexity of $\mathbb{V}$, we run the experiment for increasing the number of points $n$. As a reference, we also compute a high sample estimate of $J_g$ using $n = 10 \times 10^9$ points. The results are presented in Figure 4. The bound quickly tends towards the high sample estimate. However, we can observe the exponential trend of the execution time with respect to $n$, especially for $n \geq 10^5$, which highlights the need to alleviate the computational complexity of Voronoï diagrams.

**Heat diffusion and flow-in-a-pipe**    For the two last test cases, we first try to use the complete Voronoï diagram. Due to the high computational complexity, we do not use all the data points available and use $n = 2 \times 10^5$ for heat diffusion and $n = 8 \times 10^4$ for flow-in-a-pipe. In this case, it is enough to illustrate how expensive it is to compute the bounds using Voronoï diagrams. We still evaluate the bound with this limited

number of data points. Then, we use the mixed-random-grid structure and Proposition 4.7 to compute the bound. We report the upper bound and the execution time in Table 1. We can see that computing the bound is simply not affordable using the complete Voronoï diagram, with extensive execution times even without using all the data points. In contrast, using the mixed-random-grid structure and Proposition 4.7 allows us to compute the bound very quickly using all the data points.

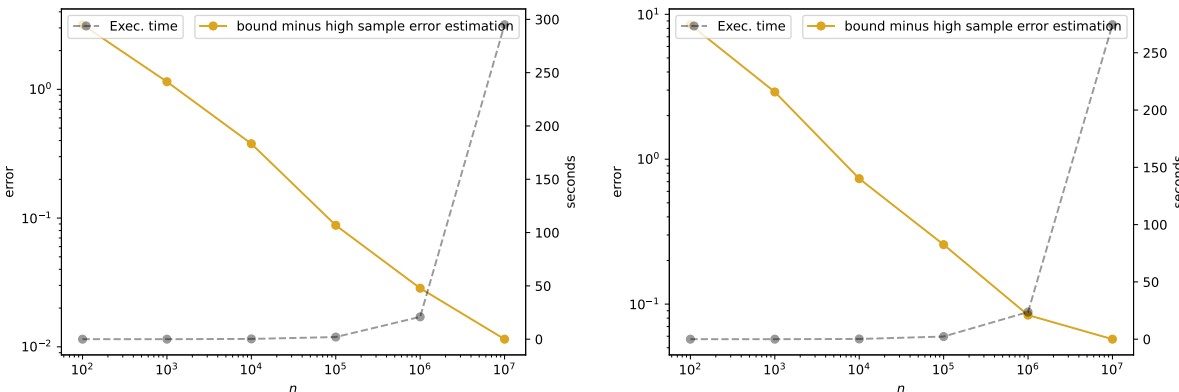

Figure 4: Evolution of the difference between the upper bound and a high sample estimation of the $J_g$ with respect to the number of points $n$ for the **Sinus function (left)** and the **Holder table function (right)**. The high sample estimation of $J_g$ is computed with $n = 10^9$. The time required for computing the bound (exec. time) is also plotted in black.

| | | Full Voronoï | | Mixed-random-grid | |
|---|---|---|---|---|---|
| | *Max. err.* | *Upper bound* | *Time (sec.)* | *Upper bound* | *Time (sec.)* |
| **Diffusion** | 0.18 | 84 ($n = 2 \times 10^4$) | 3120 | 1.67 ($n = 512 \times 10^4$) | 0.89 |
| **Flow-in-a-pipe** | 1.43 | 11.40 ($n = 8 \times 10^4$) | 7339 | 9.06 ($n = 32 \times 10^4$) | 0.007 |

Table 1: Upper bound estimation and Voronoï diagram's construction time for the heat diffusion and flow-in-a-pipe experiment.

In this section, we introduced a way to compute bounds on the generalization error of Lipschitz approximators based on Voronoï diagrams. We illustrated the computational complexity of the approach on realistic test cases and showed that leveraging a mixed-random-grid structure can dramatically reduce the computational complexity of the approach for appropriate test cases. In the next section, we build on these ideas and propose a new approach to computing the bounds by reformulating the problem into a black-box optimization problem.

## 5 Bounding as a Certified Deterministic Optimistic Optimization Problem

In this section, we introduce a new approach that involves solving a zeroth order black-box optimization problem while using principles from the previous section to ensure that the optimization result actually upper-bounds $J_g$. Let $N_k : \mathbb{X} \to \mathbb{X}$ be the $k$ nearest neighbors map built on $\{x_i\}_{i=1}^n$. Throughout this section, we focus on the following function:

$$\bar{e} : x \mapsto \min_{x_i \in N_k(x)} \left\{ |g(x) - f(x_i)| + K_f \|x - x_i\| \right\}, \tag{7}$$

where $k$ is a hyperparameter that controls the complexity of the evaluation of $\bar{e}$ – ideally, with an infinite evaluation budget, we would choose $k = n - 1$. This formulation heavily relies on the approach of Piyavskii (1972) and Shubert (1972). We will then use the following Proposition 5.1, which emphasizes how solving a black-box optimization problem will achieve the desired bound:

**Proposition 5.1.** *Let $e$ and $\bar{e}$ be defined as in Proposition 3.3 and equation 7. Then,*

$$\forall x \in \mathbb{X}, \ e(x) \leq \bar{e}(x).$$

*Therefore, we have that*

$$J_g \leq \max_{x \in \mathbb{X}} \bar{e}(x) = \max_{x \in \mathbb{X}} \min_{x_i \in N_k(x)} \big\{ |g(x) - f(x_i)| + K_f \|x - x_i\| \big\}. \tag{8}$$

The proof is left in Appendix A.3. The corresponding zeroth order black-box optimization problem is non-convex. Hence, many classical maximization techniques (e.g., bayesian optimization, CMA-ES, evolutionary algorithms, etc..) are not appropriate in our case since we want to make sure that the result of the optimization $x_*$ verifies $J_g \leq \bar{e}(x_*)$. Specifically, we must leverage the Lipschitz property of $g$ and $f$ to cope with this constraint and use certified optimization techniques. To the best of our knowledge, only one such algorithm has been explored in Bachoc et al. (2021), where the authors introduce Certified Deterministic Optimistic Optimization (c-DOO). As we shall see, this algorithm uses principles from the previous section to provide a certificate $\epsilon$ along with the maximum found at the end of the optimization, which ensures that $\max_{x \in \mathbb{X}} \bar{e}(x) \leq \bar{e}(x_*) + \epsilon$.

In this section, we explore the use of c-DOO to achieve the desired bound. We also derive the certified version of the recently introduced Voronoï Optimistic Optimization using Proposition 4.1, with a partition chosen as a Voronoï diagram.

### 5.1 Certified Deterministic Optimistic Optimization

DOO Munos (2011) is part of a line of bandit algorithms such as more recent LIPO Malherbe & Vayatis (2017), HOO Bubeck et al. (2011), or Zooming algorithm Kleinberg et al. (2008). They all come with guarantees in terms of sample complexity, e.g. the number of iterations needed to output an $\epsilon$-optimal solution that verifies $\max_{x \in \mathbb{X}} \bar{e}(x) \leq \bar{e}(x_*) + \epsilon$ but we have no guarantees that for a given $n$, this assumption will be verified. The only algorithm providing certificates along with a given maximize $x_*$ is the certified version of DOO, c-DOO Bachoc et al. (2021).

Before diving into the details of DOO and c-DOO, let's first introduce some mathematical objects we will rely on. Let's consider an infinite set $\{\mathbb{S}_{i,h}\}_{i \in \{1,\dots,n_0 s^h\}, h \in \mathbb{N}}$, once again called cells, where $s, n_0 \in \mathbb{N}^*$. For each $h$, $\mathbb{X} = \bigcup_{i=1}^{s^h} \mathbb{S}_{i,h}$, $\bigcap_{i=1}^{s^h} \mathbb{S}_{i,h} = \varnothing$. The sequence is structured such that for any $h \in \mathbb{N}$ and $l \in \{1,\dots,n_0 s^h\}$, there exists $\{i_1,\dots,i_s\} \subset \{1,\dots,n_0 s^{h+1}\}$ such that $\{\mathbb{S}_{i_1,h+1},\dots,\mathbb{S}_{i_s,h+1}\}$ form a partition of $\mathbb{S}_{i,h}$. We call $\mathbb{S}_{i,h}$ the parent subspace of $\{\mathbb{S}_{i_1,h+1},\dots,\mathbb{S}_{i_s,h+1}\}$, which we call the children subspaces of $\mathbb{S}_{i,h}$. This structure can be seen as a tree-structured subdivision of $\mathbb{X}$ parametrized by $n_0$ and $s$, where $s$ controls the number of children subspaces for each parent subspace, and $n_0$ controls the number of initial parent subspaces (for which $h = 0$). Each cell $\mathbb{S}_{i,h}$ is assigned a center $x_{i,h}$. For each cell $\mathbb{S}_{i,h}$, it is possible to bound $\max_{x \in \mathbb{S}_{i,h}} \bar{e}(x)$ as follows:

$$\max_{x \in \mathbb{S}_{i,h}} \bar{e}(x) \leq \bar{e}(x_{i,h}) + (K_f + K_g) r(x_{i,h}) \tag{9}$$

where we used equation 2 and Proposition 3.3, knowing $\bar{e}$ is $(K_f + K_g)$-Lipschitz. The quantity $r(x_{i,h})$ corresponds to the radius of the cell as defined in 4.

In our case, we define $\{\mathbb{S}_{i,0}\}_{i \in \{1,\dots,n_0\}}$ as a grid-like partition of the space already used in Section 4.1. We split each $\mathbb{S}_{i,h}$, which are hyperrectangles, into $s = 2^d$ children.

DOO and c-DOO assume that we can have access to the evaluation of the function to optimize at any $x \in \mathbb{X}$. That was not possible in our case when we were considering $e(x)$, but it is now that we focus on $\bar{e}(x)$. As an important practical consequence, we can now choose any partition of $\mathbb{X}$ as a basis for c-DOO. Hence, for each $h$, we will consider $\{x_{i,h}\}_{i=1}^{n_0 s^h}$ as grid-structured data points and split each $\mathbb{S}_{i,h}$, which are hyperrectangles, into $s = 2^d$. As a consequence, we do not have to compute $r(x_{i,h})$ in equation 9. Indeed, we have that

$$\forall h \in \mathbb{N}, \ r(x_{1,h}) = \cdots = r(x_{n_0 s^h, h}) = \delta_h, \tag{10}$$

where $\delta_h$ is the half diagonal of $\{\mathbb{S}_{i,h}\}_{i=0}^{n_0 s^h}$. If we consider $\{x_{i,0}\}_{i=1}^{n_0 s^0}$ as grid-structured data points of parameter $\{p_j\}_{j \in \{1,\dots d\}}$, then $\delta_h = \sqrt{\sum_{l=1}^d \frac{1}{2^h (p_l-1)^2}}$.

We now have all the elements needed to state the c-DOO algorithm adapted to our setting (Algorithm 1). Since $\bar{e}$ is not costly to evaluate – the computational cost mainly comes from the $k$-nearest neighbors – we initialize the algorithm with a large set of $n_0$ grid-structured data points, which will be the initial centers. Then, we build a set $\mathcal{S}$ of pairs of indices identifying current celles to consider. At each step, (**line 2**) we select the cell $(i^*, h^*)$ that maximizes the bound in equation 9, (**line 3**) compute the children cells of $\mathbb{S}_{i^*,h^*}$, which will be the new centers (for which $h = h^* + 1$). Then, (**line 4**) we compute these new centers, (**line 5**) remove the center corresponding to $(i^*, h^*)$, and (**line 6**) merge the set of indices pairs. After $T$ steps, we output the maximum bound found. An important property of this algorithm is that regardless of $T$, **the bound is always valid; it will only be tighter with a larger $T$.**

---

**Algorithm 1** Certified DOO (c-DOO) for upper bounding $J_g$ (adapted from Bachoc et al. (2021))

---

**Initialization**: Define $\bar{e}$ by selecting $k$ (see equation 7)
An initial set of $n_0$ grid-structured centers $\{x_{i,0}\}_{i=1}^{n_0}$
The values of $\bar{e}$ at the centers, $\{\bar{e}(x_{i,0})\}_{i=1}^{n_0}$
A set of pairs of indices identifying partitions $\mathcal{S} = \{(i,0)\}_{i=1}^{n_0}$
**Parameter**: desired accuracy $\epsilon$ or maximum number of steps $T$

1: **while** $(K_f + K_g)\delta_h > \epsilon$ or number of steps $< T$ **do**
2: $\quad (i^*, h^*) = \arg\max_{i,h \in \mathcal{S}} \{\bar{e}(x_{i,h}) + (K_f + K_g)\delta_h\}$
3: $\quad$ Define $\mathcal{S}^*$ as the set indices pairs identifying children cells of $\mathbb{S}_{i^*,h^*}$
4: $\quad$ Compute their centers $\{x_{i,h}\}_{(i,h) \in \mathcal{S}^*}$.
5: $\quad$ Remove $(i^*, h^*)$ from $\mathcal{S}$
6: $\quad \mathcal{S} = \mathcal{S} \cup \mathcal{S}^*$
7: **end while**
**Output**: $J_g \leq \bar{e}(x_{i^*,h^*}) + (K_f + K_g)\delta_{h^*}$

---

In practice, to speed up the convergence, we introduce another parameter: batch size $b$. At each step, we select the $b$ cells that maximize the bound in equation 9 and compute their children. It allows us to explore the space more efficiently and to converge faster. We present the results of the c-DOO algorithm on the four test cases in Section 5.3.

## 5.2 Certified Voronoï Optimistic Optimization

Voronoi Optimistic Optimization (VOO) Kim et al. (2020) is another algorithm that belongs to the same line as DOO. Instead of using grid-structured points and partitioning the space into a sequence of hyper-rectangular cells that have a parent-children property like in DOO, it randomly samples points $\{x_i\}_{i=1}^n$ and defines the cells as Voronoï cells. It then samples a new point $x_{n+1}$ to evaluate with $\bar{e}$ inside the cell $\mathbb{V}_{i^*}$ where $i^* = \arg\max_{i \in \{1,\dots,n\}} \bar{e}(x_i)$. The Voronoï diagram is updated, and the process is repeated.

With all the material presented previously, we introduce a certified version of VOO, c-VOO, using principles from the previous sections. Inspired by c-DOO, for each cell $\mathbb{V}_i$, it is possible to bound $\max_{x \in \mathbb{V}_i} \bar{e}(x)$ as follows:

$$\max_{x \in \mathbb{V}_i} \bar{e}(x) \leq \bar{e}(x_i) + (K_f + K_g)r(x_i), \tag{11}$$

which demonstrates the validity of the bound obtained at the end of the algorithm. The entire algorithm can then be stated (Algorithm 2). First, we initialize the algorithm by randomly sampling $n$ points and computing the $\bar{e}$ values at these points. Then, at each step, (**line 2**) we compute the Voronoï diagram of the points, (**line 3**) select the center $x_{i^*}$ that maximizes the bound in equation 11, (**line 4**) sample a new point $x^*$ in the Voronoï cell $\mathbb{V}_{i^*}$ – in practice, we sample a point from a uniform distribution of range $2r(x^*)$ centered at $x^*$, with rejection if it falls outside of the corresponding Voronoï cell –, and (**line 5**) add it to

the set of points. After $T$ steps, we output the maximum bound found. Like for c-DOO, we also add a parameter $b$ (not represented in Algorithm 2 for clarity), corresponding to the number of points to sample in the Voronoï diagram at each step.

---

**Algorithm 2** Certified VOO (c-VOO) for upper bounding $J_g$

---

**Initialization**: Define $\bar{e}$ by selecting $k$ (see equation 7)
An initial set of $n$ randomly sampled centers $\mathcal{X} = \{x_i\}_{i=1}^{n}$
The values of $\bar{e}$ at the centers, $\{\bar{e}(x_i)\}_{i=1}^{n}$
**Parameter**: desired accuracy $\epsilon$ or maximum number of steps $T$

1: **while** $(K_f + K_g)r(x) > \epsilon$ or number of steps $< T$ **do**
2:     Compute $\mathbb{V}$, the Voronoï diagram of $\mathcal{X}$.
3:     $i^* = \underset{i \in \{1,...,|\mathcal{X}|\}}{\arg\max} \left\{\bar{e}(x_i) + (K_f + K_g)r(x_i)\right\}$ // $r(x_i)$ computed using equation 4
4:     Sample $x_{i+1}$ such that $x_{i+1} \in \mathbb{V}_{i^*}$.
5:     $\mathcal{X} = \mathcal{X} \cup \{x_{i+1}\}$
6: **end while**
**Output**: $J_g \leq \bar{e}(x_{i^*}) + (K_f + K_g)r(x_{i^*})$

---

### 5.3 Experiments

We now test and compare c-DOO and c-VOO on the same test cases as in Section 4.4. We use the same neural networks and the same Lipschitz constants. We test c-DOO on every test case but, unfortunately, cannot afford c-VOO for the heat diffusion and flow-in-a-pipe test cases because, as in Section 4.4, the Voronoï diagram is too expensive to compute. We run the experiments with $\bar{e}$ defined using a $k$-nearest neighbor with $k = 2048$, an accuracy of $\epsilon = 10^{-3}$, and a batch size $b = 100$. We first present results for the sinus and Holder table functions to be able to discuss the comparison between c-DOO and c-VOO. Then, we present the results for the heat diffusion and flow-in-a-pipe test cases.

For the sinus and Holder table functions, we experiment with an increasing number of learning points $n$, similar to Section 4.4. For DOO, we build an initial set of $n_0$ grid-structured centers $\{x_{i,0}\}_{i=1}^{n_0}$ with parameter $\{\lfloor\sqrt{n}\rfloor, \lfloor\sqrt{n}\rfloor\}$ ($n_0 = \lfloor\sqrt{n}\rfloor^2$). For c-DOO, we choose $n_0 = n$. Note that for each $n$, the algorithm will ultimately perform $n_0 + bT$ evaluations of $\bar{e}$. The evolution of the upper bound with respect to $n$ is shown in Figure 5. On the leftmost plots, we also plot the evolution of the error obtained with Voronoï diagrams. We can see that c-DOO and c-VOO consistently achieve tighter bounds than the Voronoi approach. We also report the number of iterations with respect to $n$ until convergence, and the time per iteration. These plots emphasize the advantages of c-DOO over c-VOO, both in terms of the number of iterations and time per iteration.

For the heat diffusion and flow-in-a-pipe test cases, we perform the DOO with an initial set of $n_0$ grid-structured centers $\{x_{i,0}\}_{i=1}^{n_0}$ with parameter $\{5, 5, 5, 5, 155, 155\}$ and $\{31, 23, 155, 245\}$ respectively, to make hyperrectangles as close to hypercubes as possible. The results are plotted in Figure 6, and the best bounds found are reported in Table 2. We also report the bounds of Table 1 to ease comparison. The results are consistently better than the Mixed-random-grid version, even from the first iteration. Note that we can stop the algorithm at any time before convergence, and the obtained upper bound will be valid. Adding iteration only improves the tightness of the bound.

| | Max. abs. error | c-DOO | | Mixed-random-grid |
| --- | --- | --- | --- | --- |
| | | Upper bound | Time per iteration (sec.) | Upper bound |
| **Diffusion** | 0.18 | 0.92 ($n = 512 \times 10^4$) | 4.87 | 1.67 |
| **Flow-in-a-pipe** | 1.43 | 4.51 ($n = 32 \times 10^4$) | 1.73 | 9.06 |

Table 2: Upper bound estimation and time per iteration of c-DOO for the heat diffusion and flow-in-a-pipe experiments. The Maximum absolute error over the dataset is reported, as well as the error obtained with Mixed-random-grid presented in Table 1 for comparison.

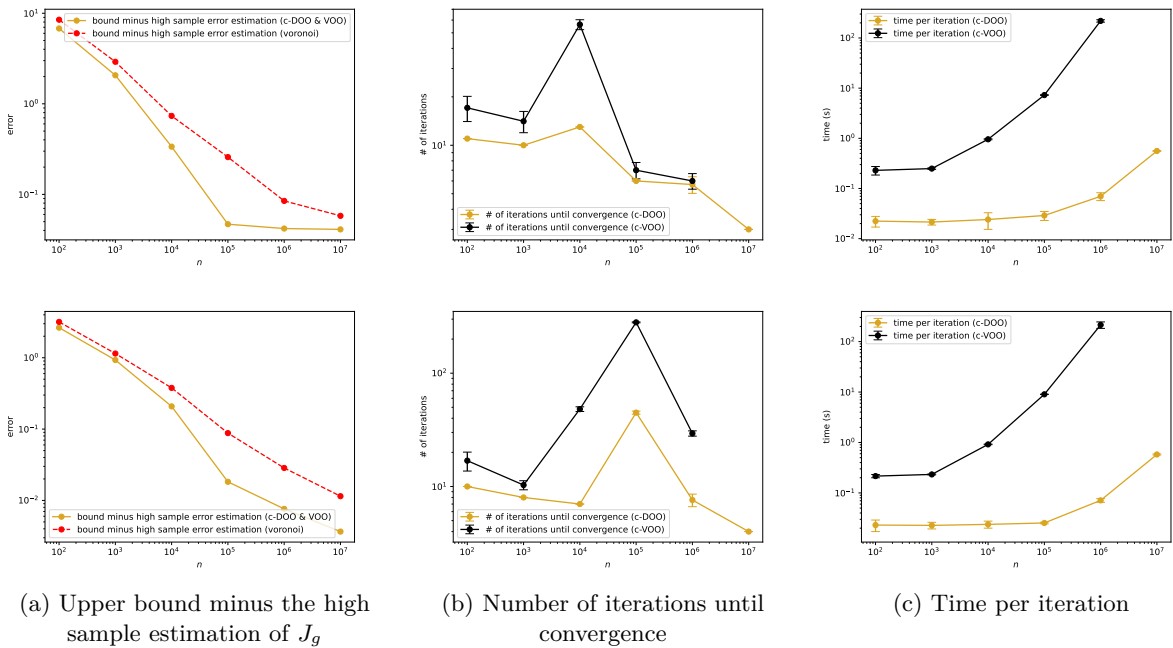

(a) Upper bound minus the high sample estimation of $J_g$

(b) Number of iterations until convergence

(c) Time per iteration

Figure 5: Comparison of error, iterations, and time per iteration for Holder **(top row)** and Sinus **(bottom row)**. The left column shows the difference between the upper bound and a high sample estimation of $J_g$. Only c-DOO is plotted because since the bound is computed with $\epsilon = 10^{-3}$, the two curves are almost indistinguishable. We also report the evolution of the error obtained with Voronoi diagrams in red. The middle column shows the number of iterations until convergence. The right column shows the time per iteration. Because of the high execution time, we did not perform the computation for c-VOO with $= 10^7$ points.

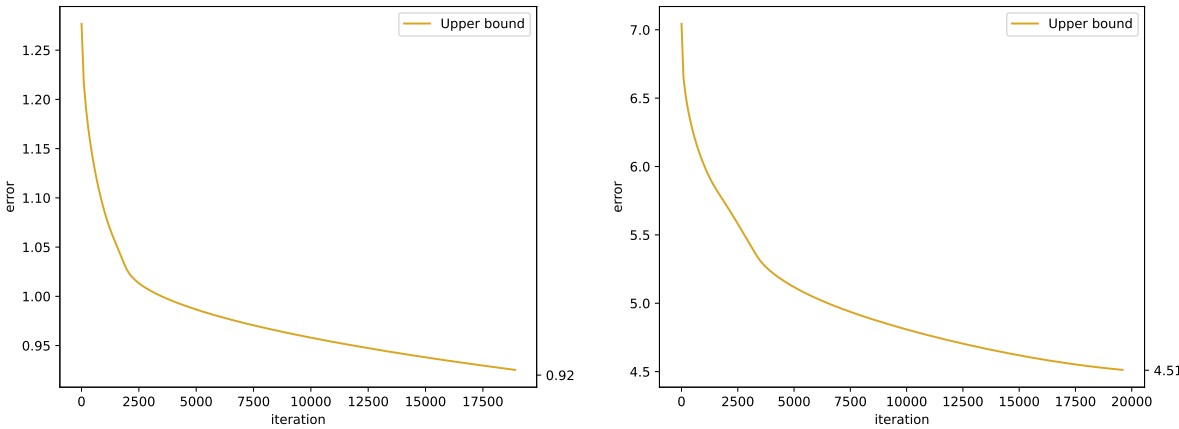

Figure 6: Evolution of the upper bound with respect to the number of iterations of c-DOO for the **heat diffusion (left)** and the **flow-in-a-pipe (right)** test cases.

## 6 Limitations and Perspectives

Before concluding, we would like to point out some limitations to our work, some ways of alleviating them, and some perspectives for future work.

**On the estimation of $K_f$**   First, computing the previous bounds requires the Lipschitz constant of $f$ to be known. Even if, in some cases, the Lipschitz constant can be known, in most cases, we can only estimate it. This calls for the question of what function $f$ is the error bound actually computed against. With the naive way of estimating $K_f$ as seen in Section 3, the estimated Lipschitz constant is a lower bound for $K_f$. However, by using this approximation, we assess the error of the Lipschitz approximator with respect to a function that does not fluctuate too much between the learning points, which is beneficial for stability guarantee purposes. Some promising works aim to estimate $K_f$ given some evaluations of $f$ in a more principled way, using extreme value theory Weng et al. (2018) and could be a good way of coping with this limitation.

**When $K_f$ is large or $f$ is not Lipschitz**   The bounds we compute might not be exploitable for functions with high $K_f$ or with discontinuities. Indeed, our bounds are linear with respect to $K_f$, so if $f$ has a high $K_f$, the bounds will be loose. This problem might be alleviated by looking at more local bounds (see below) or splitting the space into smaller regions with no discontinuities.

**The curse of dimensionality**   The curse of dimensionality is a bottleneck for the methods we presented. Indeed, the number of points needed to cover the input space $\mathbb{X}$ so that $r(x_i)$ is not too large increases with the dimension. Furthermore, the complexity of Voronoï diagrams is exponential in the dimension Aurenhammer (1991). However, moderate dimensionalities still allow Voronoï cells computations and correct space filling while corresponding to practical and modern use cases. For instance, our approach is compatible with Implicit Neural Representation Catalani et al. (2024); Serrano et al. (2023), as used with Physics Informed Neural Networks Raissi et al. (2019). Moreover, in this case, the bounds are often computed using mixed-random-grid learning points, which is common in SciML. Another way of further reducing the computational burden of the bounds would be to apply c-VOO with mixed-random-grid data in the scope of the present paper, which is a straightforward follow-up to our work. Finally, a last vein for alleviating this problem would be to optimize the computation of Voronoï cells, like in Ray et al. (2018), where the authors present a parallel algorithm for computing Voronoï cells.

## 7 Conclusion

This paper introduces novel methods to derive strict post-training generalization error bounds for Lipschitz approximators. We first leveraged the Lipschitz property and Voronoï diagrams to compute the bounds. Then, we alleviated the computational cost of computing such diagrams by taking advantage of the mesh-like structure of some data's dimensions, which is common in SciML applications where data comes from spatiotemporal meshes. Finally, we also proposed reformulating the problem as a black-box Lipschitz optimization problem, introducing c-VOO and using c-DOO algorithms to achieve tighter bounds. Our experiments on various test cases, including SciML applications, demonstrated the effectiveness of our approaches. Future work will focus on further reducing computational complexity, exploring local Lipschitz properties, and leveraging the mesh-like structure for c-VOO.

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

# A  Appendix: Proofs

## A.1  Proof of Proposition 4.1

Let's first recall Proposition 4.1:

**Proposition.** *Let's consider a partition $\{\mathbb{S}_i\}_{i=1}^n$ of $\mathbb{X}$, where each element of the partition is called a cell, such that $\mathbb{X} = \bigcup_{i=1}^n \mathbb{S}_i$, $\bigcap_{i=1}^n \mathbb{S}_i = \varnothing$ and $\forall i \in \{1,\ldots,n\}, x_i \in \mathbb{S}_i$. Let $r(x_i) = \max\limits_{x \in \mathbb{S}_i} \|x - x_i\|$. Then,*

$$J_g \leq \max_{i \in \{1,\ldots,n\}} \big\{ e(x_i) + (K_f + K_g) r(x_i) \big\}.$$

*Proof.* Let's consider $x \in \mathbb{X}$. By definition of the partition, $\exists i \in \{1,\ldots,n\}$ such that $x \in \mathbb{S}_i$. Then, by Proposition 3.3, we have

$$e(x) \leq e(x_i) + (K_f + K_g)\|x - x_i\|.$$

Since by definition of $r(x_i)$, $\forall x \in \mathbb{S}_i$, $\|x - x_i\| \leq r(x_i)$, we also have that $\forall x \in \mathbb{S}_i$,

$$e(x) \leq e(x_i) + (K_f + K_g)\|x - x_i\| \leq e(x_i) + (K_f + K_g) r(x_i).$$

Finally, since $\mathbb{X} = \bigcup_{i=1}^n \mathbb{S}_i$, it holds that $\forall x \in \mathbb{X}$, $e(x) \leq \max\limits_{i \in \{1,\ldots,n\}} \big\{ e(x_i) + (K_f + K_g) r(x_i) \big\}$. Considering that $J_g = \max\limits_{x \in \mathbb{X}} e(x)$ concludes the proof.

$\square$

## A.2  Proof of Proposition 4.7

Let's first recall Definition 4.6 and Proposition 4.7.

**Definition** (Mixed-random-grid structured data). *Let $\mathbb{J} \subset \{1,\ldots,d\}$ and $\mathbb{K} := \{1,\ldots,d\} \setminus \mathbb{J}$. Let $\mathbb{X}_{\mathbb{J}}$ and $\mathbb{X}_{\mathbb{K}}$ be the spaces built with the dimensions of $\mathbb{X}$ indexed by elements of $\mathbb{J}$ and $\mathbb{K}$, respectively. Let's note $x_{i,\mathbb{J}}$ and $x_{i,\mathbb{K}}$ the vectors built from $x_i$ by selecting its dimensions indexed by elements of $\mathbb{J}$ and $\mathbb{K}$. We say that data $\{x_i\}_{i=1}^n$ is structured as mixed-random-grid when $\{x_{i,\mathbb{J}}\}_{i=1}^n$ is structured as a grid and $\{x_{i,\mathbb{K}}\}_{i=1}^n$ is randomly sampled from any distribution $P_x$ on $\mathbb{X}_{\mathbb{K}}$.*

**Proposition.** *Let $\{x_i\}_{i=1}^n$ be structured as a Mixed-random-grid as defined in Definition 4.6, with a grid of parameters $\{p_j\}_{j \in \mathbb{J}}$. Let $\{\mathbb{V}_{i,\mathbb{K}}\}_{i=1}^n$ be the Voronoï diagram of $\{x_{i,\mathbb{K}}\}_{i=1}^n$. Then,*

$$r(x_i) = \sqrt{r(x_{i,\mathbb{K}})^2 + \sum_{j \in \mathbb{J}} \frac{1}{4(p_j - 1)^2}}.$$

*Proof.* Let's consider $x \in \mathbb{X}$. By definition of the mixed-random-grid structure, we can decompose $x$ into $x_{\mathbb{J}}$ and $x_{\mathbb{K}}$. For each $x_i$, we have $x_{i,\mathbb{J}}$ structured as a grid and $x_{i,\mathbb{K}}$ randomly sampled.

By the definition of $r(x_i)$, we have:

$$r(x_i)^2 = \max_{x \in \mathbb{S}_i} \|x - x_i\|^2.$$

We can decompose this into:

$$r(x_i)^2 = \max_{x \in \mathbb{S}_i} \big( \|x_{\mathbb{J}} - x_{i,\mathbb{J}}\|^2 + \|x_{\mathbb{K}} - x_{i,\mathbb{K}}\|^2 \big).$$

Since $\{x_{i,\mathbb{J}}\}_{i=1}^n$ is structured as a grid, the maximum distance between any $x_{\mathbb{J}}$ and $x_{i,\mathbb{J}}$ can be expressed as:

$$\max_{x \in \mathbb{S}_i} \|x_{\mathbb{J}} - x_{i,\mathbb{J}}\|^2 = \sum_{j \in \mathbb{J}} \frac{1}{4(p_j - 1)^2}.$$

For the randomly sampled part $\{x_{i,\mathbb{K}}\}_{i=1}^n$, the maximum distance is obtained using the Voronoï diagram:

$$\max_{x \in \mathbb{S}_i} \|x_\mathbb{K} - x_{i,\mathbb{K}}\|^2 = r(x_{i,\mathbb{K}})^2.$$

Combining these results, we get:

$$r(x_i) = \sqrt{r(x_{i,\mathbb{K}})^2 + \sum_{j \in \mathbb{J}} \frac{1}{4(p_j - 1)^2}}.$$

This concludes the proof. $\qquad\square$

### A.3 Proof of Proposition 5.1

Let's first recall Proposition 5.1.

**Proposition** (expanded). *Let $N_k : \mathbb{X} \to \mathbb{X}^k$ be the $k$ nearest neighbors map built on $\{x_i\}_{i=1}^n$. Let's define*

$$\bar{e} : x \to \min_{x_i \in N_k(x)} \left\{ |g(x) - f(x_i)| + K_f \|x - x_i\| \right\},$$

*where $k$ is a hyperparameter that controls the complexity of the evaluation of $\bar{e}$. Then,*

$$\forall x \in \mathbb{X}, \ e(x) \le \bar{e}(x).$$

*Therefore, we have that*

$$J_g \le \max_{x \in \mathbb{X}} \bar{e}(x) = \max_{x \in \mathbb{X}} \min_{x_i \in N_k(x)} \left\{ |g(x) - f(x_i)| + K_f \|x - x_i\| \right\}.$$

*Proof.* Since $f$ is $K_f$-Lipschitz, we have that $\forall x \in \mathbb{X}$ and $\forall i \in \{1, \ldots, n\}$

$$f(x_i) - K_f \|x - x_i\| \le f(x) \le f(x_i) + K_f \|x - x_i\|.$$

If $g(x) \le f(x_i)$,

$$e(x) = |g(x) - f(x)| = f(x) - g(x) \le f(x_i) - g(x) + K_f \|x - x_i\|,$$
$$e(x) \le |g(x) - f(x_i)| + K_f \|x - x_i\|.$$

If $g(x) \ge f(x_i)$,

$$e(x) = |g(x) - f(x)| = g(x) - f(x) \le g(x) - f(x_i) + K_f \|x - x_i\|,$$
$$e(x) \le |g(x) - f(x_i)| + K_f \|x - x_i\|.$$

In summary, $\forall x \in \mathbb{X}$ and $\forall i \in \{1, \ldots, n\}$,

$$e(x) \le |g(x) - f(x_i)| + K_f \|x - x_i\|.$$

This inequality holds $\forall i \in \{1, \ldots, n\}$ so since $\bar{e}(x) = \min_{x_i \in N_k(x)} \left\{ |g(x) - f(x_i)| + K_f \|x - x_i\| \right\}$,

$$\forall x \in \mathbb{X}, \ e(x) \le \bar{e}(x).$$

Finally, since $J_g = \max_{x \in \mathbb{X}} e(x)$,

$$J_g \le \max_{x \in \mathbb{X}} \bar{e}(x) = \max_{x \in \mathbb{X}} \min_{x_i \in N_k(x)} \left\{ |g(x) - f(x_i)| + K_f \|x - x_i\| \right\},$$

which concludes the proof. $\qquad\square$

# B   Appendix: Test Cases and Model Definition

In this section, we provide technical details for our work.

## B.1   Lipschitz neural network implementation and training

For all the test cases, the Lipschitz neural networks are built using the library REF, with 1-Lipschitz orthogonal layers and then a classical linear layer. The Lipschitz constant of the neural network is obtained by computing the highest eigenvalue of the last layer's weight matrix. The network is trained using the Adam optimizer with a learning rate schedule of $\{10^{-i}\}_{i=1}^{5}$ and a batch size of 128. The networks are trained for 1000 epochs with early stopping.

## B.2   Sinus and Holder Table Functions

Sinus and Holder table functions are both defined on the domain $[-5,5]^2$. We sample $n = 10^4$ points $\{x_i\}_{i=1}^{n}$ uniformly in this domain, evaluate the function $f$, and then rescale the domain to $[0,1]$. We then train a Lipschitz neural network on the obtained dataset. As a result, the actual function $f$ is a little bit different from the initial function because of the rescaling.

**Sinus function**   Before the rescaling, the Sinus function is defined as:

$$f : x, y \rightarrow \sin(x)\sin(y), \; x, y \in [-5,5]^2,$$

and after the rescaling, it is defined as:

$$f : x, y \rightarrow \sin(10x - 5)\sin(10y - 5), \; x, y \in [0,1]^2.$$

Its lipschitz constant is $K_f = 10$.

**Holder function**   Before the rescaling, the Holder function is defined as:

$$f : x, y \rightarrow \left| sin(x)cos(y) \exp\left( \left| 1 - \frac{\sqrt{x^2 + y^2}}{\pi} \right| \right) \right|, \; x, y \in [-5,5]^2,$$

and after the rescaling, it is defined as:

$$f : x, y \rightarrow \left| sin(10x - 5)cos(10y - 5) \exp\left( \left| 1 - \frac{\sqrt{(10x - 5)^2 + (10y - 5)^2}}{\pi} \right| \right) \right|, \; x, y \in [0,1]^2$$

Its lipschitz constant is $K_f = 10$.

## B.3   Heat Diffusion

$f : \mathbb{X} \subset \mathbb{R}^6 \rightarrow \mathbb{R}$ as the stationary solution of the heat equation in two dimensions with 4 Dirichlet boundary conditions. The heat equation is defined as:

$$\begin{cases} \dfrac{\partial f}{\partial t} = D \left( \dfrac{\partial^2 f}{\partial x^2} + \dfrac{\partial^2 f}{\partial y^2} \right) \\ f(x, 0, t) = b_1, f(x, 32, t) = b_2, f(0, y, t) = b_3, f(32, y, t) = b_4 \end{cases}$$

The final solution depends on the boundary conditions but not on the initial state.

The 6 dimensions in $\mathbb{X}$ correspond to the two spatial dimensions and the 4 boundary conditions. The function $f$ returns the temperature over the domain. Specifically, for each set of boundary condition $\{b_1, b_2, b_3, b_4\} \in [0,1]^4$, we run the simulation on a $32 \times 32$ grid $(x, y \in [0,31]^2)$ and collect the value of the temperature at each cell of the grid, $\{T_{i,j}\}_{i,j=1}^{32}$, resulting in $32 \times 32$ learning points per simulation. We run 5000 simulations for a total of $n = 512 \times 10^4$ learning points.

The simulations are run using py-pde Zwicker (2020).

### B.4 Flow in a Pipe

$f : \mathbb{X} \subset \mathbb{R}^4 \to \mathbb{R}$ as the stationary solution of a vicious flow in a pipe. It is governed by the equation:

$$\begin{cases} \dfrac{\partial \mathbf{v}}{\partial t} + (\mathbf{v} \cdot \nabla)\mathbf{v} = -\nabla p + \nu \Delta \mathbf{v}, \\ \nabla \cdot \mathbf{v} = 0, \\ \mathbf{v}(x, 0, t) = (0, 0), \\ \mathbf{v}(x, 32, t) = (0, 0), \\ \mathbf{v}(0, y, t) = (v_0, 0). \end{cases}$$

The 4 dimensions in $\mathbb{X}$ correspond to the two spatial dimensions and the two boundary conditions. The function $f$ returns the pressure and speed along $x$ and $y$ over the domain, but we only consider the speed along $x$. Specifically, for each set of parameters $\{v_0, \nu\} \in [0.01, 1] \times [0.01, 5]$, we run the simulation on a $50 \times 32$ grid ($x, y \in [0, 31] \times [0, 50]$) and collect the value of the temperature at each cell of the grid, $\{v_{i,j}\}_{i,j=1}^{50,32}$, resulting in $50 \times 32$ learning points per simulation. We run 2000 simulations for a total of $n = 320 \times 10^4$ learning points.

The simulations are run using phi-flow Holl et al. (2020).

