# OpenReview forum: "Instance-dependent Approximation Guarantees for Lipschitz Approximators, Application to Scientific Machine Learning"
_TMLR — Rejected by TMLR_

### Review · Reviewer_bhTs · 2025-02-19

**Summary Of Contributions:**

The authors present a way to establish an upper bound on the generalization error $\|f-g\|_\infty$ assuming the Lipschitz constant of both $f$ and $g$. To do this, they rely on a set of base points and the evaluation of the error on these points, divide the full space into the neighborhood of base points (high-dimensional Vonoroi diagram), and then use the fact that the generalization error function is Lipschitz to derive a sound upper bound. Some simple case studies are presented.

**Audience:**

Yes

**Broader Impact Concerns:**

The paper does not include a broader impact section, but I do not identify concerns.

**Claims And Evidence:**

Yes

**Requested Changes:**

1. Complete reference. In the related work, the authors mentioned some early works about robustness certification, but missed a large body of related literature. For example, they should at least cite [1-13] on robustness certification and [14-17] on using Lipschitz networks. This is not a complete list of related works, and the authors should discuss more.

2. Method based on the motivation. The authors suggested to use sampling to obtain an approximation of the Lipschitz constant of both the target function $f$ and the approximator $g$. However, it seems directly evaluating the Lipschitz constant of $f-g$ does not incur additional overhead, and is a much better approximation to the true Lipschitzness than $K_f + K_g$. Why do the authors then choose to approximate $K_f$ and $K_g$ instead? Their method can then be naively substituted with the new Lipschitz constant, which is expected to be much smaller. For example, assume $f(x) = g(x) =x$, then $K_f + K_g =2$ but $K_{f-g} = 0$. Sampling in this case results in the true Lipschitz constant but has a large error. A discussion should be included.


**Reference**

[1] On the effectiveness of interval bound propagation for training verifiably robust models.

[2] An abstract domain for certifying neural networks.

[3] Scaling provable adversarial defenses.

[4] General cutting planes for bound-propagation-based neural network verification.

[5] Understanding certified training with interval bound propagation.

[6] Complete verification via multi-neuron relaxation guided branch-and-bound.

[7] Automatic perturbation analysis for scalable certified robustness and beyond.

[8] CTBENCH: A Library and Benchmark for Certified Training.

[9] Certified training: Small boxes are all you need.

[10] Expressive losses for verified robustness via convex combinations.

[11] Gaussian Loss Smoothing Enables Certified Training with Tight Convex Relaxations.

[12] EXPRESSIVITY OF RELU-NETWORKS UNDER CONVEX RELAXATIONS.

[13] Connecting Certified and Adversarial Training.

[14] CLIP: Cheap Lipschitz Training of Neural Networks

[15] Training Certifiably Robust Neural Networks with Efficient Local Lipschitz Bounds

[16] Training robust neural networks using Lipschitz bounds

[17] Sorting Out Lipschitz Function Approximation

**Strengths And Weaknesses:**

Strength: the method is sound, presenting a sound upper bound of the generalization error. Presentation is clear, and method is well-motivated.

Weakness: using the same motivation, it seems the method can be greatly simplified and improved. See details below.

---

> ### Author Response · Authors · 2025-04-11
>
> We would like to thank reviewer bhTs for recognizing the soundness and clarity of our work.
>
> We concatenate the answer to the reported weakness with bullet point 2 below.
>
> ### Requested change
>
> 1. We initially did not want to dig deep into the literature on formal verification and Lipschitz optimization and only provided the main papers of this field for clarity and conciseness of the positioning (let's just point out that we had already cited [17], which we agree is seminal). But we acknowledge that this might not be sufficient. We thank you for the provided references, which we complemented our related work with. You can find it in the revised manuscript.
>
> 2.  We thank you for your suggestion. We agree that the real Lipschitz constant of $f-g$ is probably lower than $K_f + K_g$ in practice. However, as we discuss in Section 3.2, estimating the Lipschitz constant based on samples is prone to estimation error. Allow us to recall here that unlike what you reported, **we do not need to estimate $K_g$ because we know it** thanks to the use of Norm-preserving neural nets. We could estimate $K_g$ as you suggest, regardless, but we prefer not to do this to mitigate the estimation error as much as possible. In addition, we deem the estimation error of $K_f$ less problematic than that of $K_g$, for the reasons detailed in the paragraph "On the estimation of $K_f$ of Section 6. We would be interested in knowing your opinion after this clarification.

---

> > ### Comment · Reviewer_bhTs · 2025-04-14
> >
> > Dear AC and authors,
> >
> > I think the reply from the authors clears my concerns.

---

### Review · Reviewer_2Akm · 2025-04-01

**Summary Of Contributions:**

The paper studies the problem of deriving instance-dependent generalization error bounds for models that are Lipschitz continuous. The motivation of this comes from scientific machine learning (SciML) applications. For low-dimensional problems the authors are able to derive upper bounds using geometric/sampling techniques. The authors also show how to get tighter error bounds using a version of deterministic optimistic optimization (DOO) previously introduced in Bachoc et al (2021).

**Audience:**

No

**Claims And Evidence:**

No

**Requested Changes:**

Major changes:
* Consider strengthening the results adnd connecting more tightly to scientific machine learning problems. For example, are there special cases of the Heat equation or flow-in-a-pipe that enables you to get better upper bounds than the generic ones derived in Section 4?
* Consider engaging with the SciML literature on error bounds better.

Other comments:
* I would clarify the statement that "neural networks...naturally enjoy the Lipschitz property". Networks that use self-attention (such as transformers) do not satisfy this. Cf "The Lipschitz constant of self attention", ICML 2021.
* It is not clear how Figure 1 illustrates Prop 4.1. Please add an explanation.
* The mixed-random-grid setting seems a bit artificial. Has this been introduced in a previous paper? The authors cite 4 papers in the beginning of Section 4.3; which of these papers studies this setting? In any case I don't see how you get too much speedup; the runtime continues to be exponential in d?
* Algorithm 1 seems to be derived from a previous paper, so consider citing it in the description/caption of the algorithm.
* I am not sure how to interpret Figure 5a (left column).
* I am not sure why Figure 5b (middle) column has curves that are not monotonic.
* I don't know how to interpret Figure 6. Why are these curves interesting?

**Strengths And Weaknesses:**

**Strengths**
* The topic is interesting.
* Establishing error bounds for scientific machine learning models can be beneficial in applications.

**Weaknesses**
* The contributions of this paper are theoretical, but the theory is not strongly connected to scientific machine learning. Most of the derived error bounds are applicable for generic neural networks. The application/experimental results to SciML are also fairly weak; they could equally have been applied to other types of problems (say, in computer vision) that involve implicit neural representations.
* The paper unfortunately does not engage with the large body of work on generalization bounds for scientific machine learning models. I would consider starting from "Estimates of the generalization error of PINNs", Misra-Molinaro (2020) and working through its many references all the way up until 2024.
* Most of the theory/algorithms either is fairly straightforward, or is derived from previous papers (such as Bachoc et al 2021). While novelty is not a review criterion, clarity is; and the authors should probably do a better job conveying this more clearly.

---

> ### Author Response · Authors · 2025-04-11
>
> We would like to thank reviewer 2Akm for recognizing the value and importance of the topic we tackle.
>
> ### Weaknesses
>
> * Our theory can indeed be applied to general Lipschitz approximation. However, the Mixed-random-grid upper bound is strongly connected to the learning setting encountered in scientific machine learning because it leverages the fact that some dimensions of the learning points are tensorized (spatio-temporal coordinates - see for instance use cases of PDEBench [6]), which is rarely seen outside of this field. As for DOO variants, they are indeed applicable to other types of learning problems, but we wanted to keep our evaluation benchmark consistent over the paper. In addition, our theory requires us to assume that the function to learn, $f$ is Lipschitz, which is a reasonable assumption in SciML but not in computer vision (for instance, at the edge of objects, $f$ is discontinuous). We carefully chose the title of our paper to make it clear that our theory goes beyond SciML, but that we applied it to such benchmarks - for the previously aforementioned reasons.
>
> * This is indeed a missing part of our related work, and we sincerely thank you for pointing it out. We added a discussion in the revised manuscript. However, as we discuss, these bounds are mainly theoretical [1] [2] or leverage PDE-specific information [3] [4]. In this work, our goal is to provide bounds for any class of learning problem, regardless of the underlying PDE, and to unlock generalization bounds for use cases that encompass different instances of PDE solutions, e.g., when PDE's parameters or boundary conditions involved in the simulation change within the dataset (as illustrated in NeurIPS 2024 competition ML4CFD [5]).
>
> * We respectfully disagree on the clarity of our contribution; we clearly delimitate them and even cite the work that we build our paper on in the abstract. We make this even clearer by citing [7] in Algorithming 1, following your legitimate advice. As for the novelty, we demonstrate three properties (see appendices), adapt c-DOO to the setting of error bounding, and introduce c-VOO, which did not exist before. In the end, we derive a new class of generalization bounds that did not exist before either.
>
> ### Requested changes
>
> * As we discuss in the revised version of the paper, considering PDE-based learning problems that encompass several instances of PDEs (e.g., varying PDE parameters) makes it particularly difficult to use them for deriving generalization bounds. Since our bounds are general, we judged that it would be a sufficient and valuable contribution to the community. But this is a very interesting avenue for future contribution that we will surely look into.
> * We added a paragraph in the related works mentioning generalization error bounds specific to the scientific machine-learning field.
>
> Other comments:
>
> * You are perfectly right that vanilla self-attention is not Lipschitz and we specify this exception in the revised manuscript (Remark 3.2). Still, some works try to alleviate this by designing orthogonal attention [8].
> *  We add an explanation in the revised manuscript. This Figure illustrates that inside a partition of the space (here, 1 dimensional), we can bound the variations of $f$ and $g$ within each partition using knowledge of $K_f$ and $K_g$. Hence, we can bound $f-g$, which is illustrated by the dotted lines.
> * The mixed-random-grid has not been introduced in previous work to the best of our knowledge, but we needed to formalize it to derive Proposition 4.7. This formalism encompasses many cases of SciML, including, as an example, all the use cases of PDEBench [6]
> * We modify the title of the algorithm
> * Figure 5a presents the evolution of the error wrt the number of learning points. We display the error minus the oracle estimation of $J_g$ so that the minimum is zero, which makes the plot better readable with log axis.
> * The curves are not monotonic because there is no reason that the number of iterations required to achieve convergence increases with $n$. On the contrary, with higher $n$, the number of iterations should decrease because the partition is already dense, and so will be $\delta_0$, which is visible in the second part of the plot (from $n=10^5$ onward).
> * These curves are interesting because they show the evolution of the upper bound wrt the number of iterations. It emphasizes that even for a moderate number of iterations, the upper bound is tight. We added this comment in the revised manuscript.

---

> > ### Author Response · Authors · 2025-04-11
> > **References for official comment**
> >
> > [1] Kovachk et al. On universal approximation and error bounds
> > for fourier neural operators. Journal of Machine Learning Research 2021
> >
> > [2] Lee et al.. On the training and generalization of deep operator networks. SIAM
> > Journal on Scientific Computing, 46
> >
> > [3] Mishra et al.. Estimates on the generalization error of physics-informed neural
> > networks for approximating pdes. IMA Journal of Numerical Analysis, 2023.
> >
> > [4] De Ryck et al.. Error analysis for physics-informed neural networks (pinns) approximating
> > kolmogorov pdes. Advances in Computational Mathematics, 2022.
> >
> > [5] Bachoc et al. Instance-Dependent Bounds for Zeroth-order
> > Lipschitz Optimization with Error Certificates. NeurIPS 2021.
> >
> > [6] Takamoto et al. "Pdebench: An extensive benchmark for scientific machine learning." NeurIPS 2022
> >
> > [7] Ygoubi et al. Neurips 2024 ml4cfd
> > competition: Harnessing machine learning for computational fluid dynamics in airfoil design. In NeurIPS
> > 2024 Competition Track.
> >
> > [8] Xiao et al. "Improved Operator Learning by Orthogonal Attention." ICML 2024

---

### Review · Reviewer_aZnL · 2025-04-02

**Summary Of Contributions:**

This paper studied post-training generalization error bounds for ML models that are Lipschitz continuous. It studied the bound by partitioning the domain via the vanilla grid structured data, then introduced Voronoï diagrams, and further incorporated these two and proposed the analysis under the mixed-random-grid case. Also they recast the problem of bounding as an optimization problem, and proposed c-DOO and c-VOO to achieve tighter error bounds. The empirical results on heat diffusion and flow in a pipe illustrate the effectiveness of  these approaches in practice under moderate dimensions.

**Audience:**

Yes

**Broader Impact Concerns:**

No.

**Claims And Evidence:**

Yes

**Requested Changes:**

1. You mentioned c-DOO and c-VOO can achieve "tighter error bounds", I assume the "tigher" is compared to the approaches in Section 4, so I expect there can be an experiment comparison to further augment the claim, maybe I suggest to revise Figure 5 and 6 and add the comparison results of previous approaches to verify "tighter".
2. Is there any way (or existing research) to compare prior/post-training generalization bounds, I expect to see that these instance-dependent bounds can have lower/better bounds, so they can characterize the generalization better.

**Strengths And Weaknesses:**

Strength:
1. The motivation is clear, pursuing instance-dependent post-training generalization bound exploits the actual trained model’s structure, which should be able to better characterize the bounds in practice.
2. Novelty. The proposed approaches is based on space partitioning, which should be classical, but they further introduced Voronoï diagrams, and considered the mixed-random-grid case to adapt to higher dimension case. The extension to c-DOO and c-VOO can help to further tighter the bounds. The proposed approaches reveal certain novelty.

Weakness:
1. The theory requires Lipschitz continuity of the the groundtruth function $f$ and ML model $g$, which may be impractical, also the estimation of $K_f$ and $K_g$ is not fully discussed (as authors discussed).
2. Seem that the proposed approach only works for low or moderate dimensions in general, which may hinder the applicability of the approach.
3. The outperformance of the proposed paradigm and approaches compared to classical prior-training approaches is not fully evidented. For example can we expect the bound to be dimension-independent in theory, also it is better to have a experimental comparison if possible.

---

> ### Author Response · Authors · 2025-04-11
>
> We would like to thank reviewer aZnL for acknowledging the motivation behind our work and its novelty.
>
> ### Weakness
>
> 1. We would like to point out that **the estimation of $K_g$ is not a problem** since we use neural networks whose Lipschitz constant is enforced and thus known. As for the estimation of $K_f$, we dedicate a subsection to this topic (Section 3.2) and present potential solutions in the discussion section (Section 6, "On the estimation of $K_f$"). Could you help us improve these discussions by pointing out what you deem insufficient or too limited?
>
> 2. This work is dedicated to alleviating the curse of dimensionality of naive approach based on Voronoi (by taking into account mixed-random-grid and using DOO), making it applicable to dimensions of a range of real-world test cases. That said, we indeed acknowledge that the method still suffers from it (see dedicated discussion in section 6).
>
> 3. We do not compare our bounds to classical bounds referenced in the related work because they are usually not worst-case bounds and generally increase wrt the number and value of the neural networks. Hence, these bounds increase when the neural networks grow larger. This is detrimental because practitioners often seek to train larger networks to better fit $f$. Our bound does not depend on the neural network and only requires its Lipschitz constant, which is exactly known for the class of neural nets that we use. As a result, we can increase the size of the neural network and, therefore, its performance without affecting the tightness of our bounds.
>
> ### Requested changes
>
> 1. We added the evolution of the error of Section 4 in Figure 5 of Section 5 for holder and Sinus functions and recalled the error of Section 4 for flow-pipe and heat-diffusion in Table 2 of Section 5 to emphasize the superiority of DOO variants.
>
> 2. We derive worst-case upper bounds, i.e. bounds on the maximum error, while usual generalization bounds are either probabilistic or evaluated for $L_2$ error. Hence, we did not deem the comparison relevant to the scope of our work.

---

### Author Response · Authors · 2025-04-14

Dear AC and reviewers,

Thank you for your constructive feedback. We would like to point out that each reviewer recognized the relevance of our work to the community, and two of them acknowledged its novelty.

The main problem raised was a lack of positioning with respect to existing literature, which we addressed by complementing our related works and some other paragraphs in our revised manuscript. We also added discussions stemming from the reviewer's remarks when appropriate.

The changes are highlighted in blue in our revised manuscript.

---

> ### Comment · Reviewer_2Akm · 2025-04-14
> **Apologies for the negative feedback**
>
> But in my opinion, adding two short paragraphs, and a few citations, is not sufficiently engaging with related work. How do your bounds compare with previous SciML generalization bounds? Do you get any improvements, or not? Are there any new insights derived from your method?
>
> Thank you for clarifying that the mixed-random setting is novel. But then my concern about it being an artificial construction remains -- and this is really the core set of results around which the paper is centered.
>
> I also continue to not see how Figures 5 and 6 make sense. Figure 6 shows an upper bound, but how does this show tightness (as you claim in your response)?

---

> ### Author Response · Authors · 2025-04-14
> **Please do not apologize for engaging in a scientific debate!**
>
> We answer your comment by quoting each of your remaining concerns as sections.
>
> ### "in my opinion, adding two short paragraphs, and a few citations, is not sufficiently engaging with related work"
>
> We add some more references and additional discussions in the following response. We will include them in the revised version at the end of the discussion period.
>
> ###  "How do your bounds compare with previous SciML generalization bounds? Do you get any improvements, or not? Are there any new insights derived from your method?"
>
> The previous SciML generalization bounds cannot be compared to ours because they are either theoretical [6, 7, 8, 9] so *descriptive* or tied to a specific instance of PDE for PINNs [1, 2]. In our case, we consider several instances of a PDE in the same learning problem, i.e., different boundary conditions and PDE coefficients. Several simulations are run to obtain the same training dataset, so several **different PDEs** are solved (although of the same PDE class) within this dataset. In the heat equation test case, the temperature at the boundary is an input variable, and in the flow pipe, the upstream speed and viscosity are as well. This is not covered by [1, 2], nor PINNs in general that require fixed parameters and boundary conditions to learn PDEs solution (except with meta-learning [3], but we did not find any generalization bound for that case).
>
> In addition, contrarily to all these works, we derive a bound for the worst-case error, i.e. the supnorm over the input domain. We did not find any comparable generalization bounds for SciML or general ML literature. As a remark, in [1], the authors introduce a novel bound on generalization error but do not compare it to any generalization bound. It is not a methodological mistake but simply not relevant in their case. We argue that it is the same for us.
>
> We also argue that not using PDE-specific information is a strength rather than a weakness. Our bound is, hence, broadly applicable to any SciML learning problem where the Lipschitness assumption is relevant. The main insight derived from our work is that, before, it was not possible to provide worst-case generalization error bounds for such generic test cases, and now it is. It allows for providing workable and concrete error guarantees, which is critical e.g. when using SciML for designing safety-critical engineering systems.
>
> ### "Thank you for clarifying that the mixed-random setting is novel. But then my concern about it being an artificial construction remains -- and this is really the core set of results around which the paper is centered."
>
> We respectfully disagree that the mixed-random-grid setting is artificial. It corresponds to a formalization of the setting of all test cases of PDEBench [4]. It also formalizes the test cases of the recently published benchmark The Well [5]. Both are extensive benchmarks published at NeurIPS. Actually, it works for any uniform discretization of time and space, which is not artificial and quite common. In addition, it is not the core result around which the paper is centered. c-DOO and c-VOO do not use it (yet). However, it is our first future perspective to apply this setting to c-DOO and c-VOO because it works very well and covers a large set of test cases.
>
> ###  "I also continue to not see how Figures 5 and 6 make sense. Figure 6 shows an upper bound, but how does this show tightness (as you claim in your response)?"
>
> These Figures show that the c-DOO bounds are tighter than the Voronoi bounds. It shows how useful it can be to cast the bounding problem into an optimization problem.
> - **For Figure 5**: The curve shows the evolution of the bound for different numbers of learning points. It is interesting to see the value of the bound we can achieve wrt the number of learning points. It is like monitoring the accuracy of a Monte Carlo estimation wrt the number of sample points, for instance.
> - **For Figure 6**: When we say " even for a moderate number of iterations, the upper bound is tight", we mean that " even for a moderate number of iterations, the upper bound is tight**er than Voronoï and already close to the convergence value**". that is why the curve is interesting; like the monitoring of a validation error with the epoch, for instance.
>
> Does it make more sense after these clarifications?

---

> > ### Author Response · Authors · 2025-04-14
> > **References**
> >
> > [1] Mishra et al.. Estimates on the generalization error of physics-informed neural networks for approximating pdes. IMA Journal of Numerical Analysis, 2023.
> >
> > [2] De Ryck et al.. Error analysis for physics-informed neural networks (pinns) approximating kolmogorov pdes. Advances in Computational Mathematics, 2022.
> >
> > [3] Penwarden, Michael, et al. "A metalearning approach for physics-informed neural networks (PINNs): Application to parameterized PDEs." Journal of Computational Physics 477 (2023): 111912.
> >
> > [4] Takamoto et al. "Pdebench: An extensive benchmark for scientific machine learning." NeurIPS 2022
> >
> > [5] Ohana, Ruben, et al. "The well: a large-scale collection of diverse physics simulations for machine learning." Advances in Neural Information Processing Systems 37 (2024): 44989-45037
> >
> > [6] De Ryck, Tim, and Siddhartha Mishra. "Generic bounds on the approximation error for physics-informed (and) operator learning." Advances in Neural Information Processing Systems 35 (2022): 10945-10958.
> >
> > [7] Kovachk et al. On universal approximation and error bounds for fourier neural operators. Journal of Machine Learning Research 2021
> >
> > [8] Lee et al.. On the training and generalization of deep operator networks. SIAM Journal on Scientific Computing, 46
> >
> > [9] Benitez, Jose Antonio Lara, et al. "Out-of-distributional risk bounds for neural operators with applications to the Helmholtz equation." Journal of Computational Physics 513 (2024): 113168.

---

### Decision · Action_Editor_SYTX · 2025-04-28

**Recommendation:** Reject

**Comment:**

Although the paper offers an interesting direction and two reviewers see merit, the relevant methodological and empirical deficiencies highlighted by Reviewer 2Akm remain unresolved. Under TMLR’s evaluation policy, papers must be rejected when bold claims are not backed by sufficiently rigorous or clearly presented evidence. I therefore recommend rejection, encouraging the authors to undertake a revision that:

* Provides a comprehensive comparison with prior SciML generalization bounds, explicitly **positioning contributions and limitations**.

* **Better clarifies the experimental methodology**, especially the interpretation of Figures 5 & 6 and the tightness of the proposed bounds.

A thoroughly revised manuscript addressing these points would be welcome as a new submission.

**Audience:**

A good number of readers interested in generalization guarantees for scientific machine learning will find these results valuable, even if the work doesn’t appeal to everyone.

**Claims And Evidence:**

While two reviewers (aZnL and bhTs) considered the submission’s theoretical and empirical evidence adequate, a third reviewer (2Akm) maintained that the work still lacks (i) a thorough engagement with—and comparison to—prior SciML generalization-bound literature and (ii) a clear presentation of the numerical results (notably Figures 5 & 6). Because these relevant concerns remain unresolved after the rebuttal, the paper only partially satisfies the criterion of being “supported by accurate, convincing and clear evidence.” In consquence, further substantive revision is needed before the claims can be regarded as fully substantiated under TMLR standards.

**Resubmission Of Major Revision:**

The authors may consider submitting a major revision at a later time.